

# Anatomical location of AICA loop in CPA as a prognostic factor for ISSNHL

Sang Hyub Kim[1], Yeo Rim Ju[1], Ji Eun Choi[1], Jae Yun Jung[1], Sang Yoon Kim[2] and Min Young Lee[1]

[1] Department of Otolaryngology-Head & Neck Surgery, Dankook University Hospital, Cheonan, Chungnam, South Korea
[2] Department of Radiology, College of Medicine, Dankook University, Cheonan, Chungnam, South Korea

Corresponding author
Min Young Lee,
eyeglass@dankook.ac.kr,
eyeglass210@gmail.com

## ABSTRACT

The cerebellopontine angle (CPA) is a triangular-shaped space that lies at the junction of the pons and cerebellum. It contains cranial nerves and the anterior inferior cerebellar artery (AICA). The anatomical shape and location of the AICA is variable within the CPA and internal auditory canal (IAC). A possible etiology of idiopathic sudden sensorineural hearing loss (ISSNHL) is ischemia of the labyrinthine artery, which is a branch of the AICA. As such, the position of the AICA within the CPA and IAC may be related to the clinical development of ISSNHL. We adopted two methods to classify the anatomic position of the AICA, then analyzed whether these classifications affected the clinical features and prognosis of ISSNHL. We retrospectively reviewed patient data from January 2015 to March 2018. Two established classification methods designed by Cahvada and Gorrie et al. were used. Pure tone threshold at four different frequencies (0.5, 1, 4, and 8 kHz), at two different time points (at initial presentation and three months after treatment), were analyzed. We compared the affected and unaffected ears, and investigated whether there were any differences in hearing recovery and symptoms between the two classification types. There was no difference in AICA types between ears with and without ISSNHL. Patients who had combined symptoms such as tinnitus and vertigo did not show a different AICA distribution compared with patients who did not. There were differences in quantitative hearing improvement between AICA types, although without statistic significance ($p = 0.09$–$0.13$). At two frequencies, 1 and 4 kHz, there were differences in Chavda types between hearing improvement and no improvement ($p < 0.05$). Anatomical variances of the AICA loop position did not affect the incidence of ISSNHL or co-morbid symptoms including tinnitus and vertigo. In contrast, comparisons of hearing improvement based on Chavda type classification showed a statistical difference, with a higher proportion of Chavda type 1 showing improvements in hearing (AICA outside IAC).

Subjects Anatomy and Physiology, Neurology, Otorhinolaryngology, Radiology and Medical Imaging
Keywords AICA, CPA, Sudden hearing loss, Prognosis

## INTRODUCTION

The cerebellopontine angle (CPA) is triangular-shaped space filled with cerebrospinal fluid, and is located at the junction of the pons and cerebellum. It contains several crucial structures such as cranial nerves V to VIII, and arteries such as superior cerebellar artery

(SCA) and anterior inferior cerebellar artery (AICA). The internal auditory canal (IAC), which is a nerve canal surrounded by bone, rises anterolaterally from the CPA to reach the peripheral cochleovestibular organs. The IAC contains cranial nerves VII and VIII, which are ultimately responsible for facial muscle movement, hearing, and balance (*Rhoton Jr, 2000*).

The AICA is a branch of the basilar artery and courses through the CPA posterolaterally to supply the anterior to middle parts of the cerebellum and inferolateral pons. It branches into the labyrinthine artery, which is the sole vascular supply for the labyrinth, cochlea, and vestibular organs. The anatomical shape and location of the AICA is variable in the CPA (*Kim et al., 1990*). In postmortem and imaging studies, it has been found within the IAC in 15 to 40% of patients (*De Carpentier et al., 1996*; *Mazzoni & Hansen, 1970*; *Reisser & Schuknecht, 1991*).

Idiopathic sudden sensorineural hearing loss (ISSNHL) is a commonly seen disease in the otologic clinic. However, there is no known pathophysiology and current treatment relies on the use of systemic or intra-tympanic steroids. Possible hypotheses include inflammation, labyrinthine artery occlusion, or damage to the cochlear nerve (*Byl Jr, 1984*; *Merchant, Adams & Nadol Jr, 2005*). Given that the labyrinthine artery is a branch of the AICA, it is plausible that differences in the anatomical variation of the AICA results in the clinical findings of ISSNHL.

A number of studies have shown cases of hearing loss with an AICA located within the IAC, (*Moosa et al., 2015*) and have correlated various AICA locations with audio-vestibular symptoms (*Chadha & Weiner, 2008*; *De Carpentier et al., 1996*; *Gorrie et al., 2010*; *Kazawa, Togashi & Ito, 2013*). However, none of these studies has focused on ISSNHL, which could have a different pathophysiology given the vast differences in clinical features and diagnostic criteria. As such, in the present study, we used two previously reported methods of classification to reveal the correlation of distance of AICA loop and IAC (Chavda type) or contact of nerves and vessel (Gorrie type) to hearing status. We then analyzed whether these classifications affected the clinical features and prognosis of ISSNHL.

## MATERIALS AND METHODS

### Subjects and design

We retrospectively reviewed patient data from January 2015 to March 2018. This study was approved by the institutional review board of Dankook University Hospital (Ethical Application Ref: 2017-08-003). All patients who were admitted for ISSNHL were enrolled in the study. Verbal informed consents from participants were received. ISSNHL was diagnosed according to traditional criteria, which is defined as a threshold shift of greater than 30 dB in three consecutive frequencies, or if the patient has new hearing loss in a duration less than 3 days (*Anderson & Meyerhoff, 1983*; *Mattox & Simmons, 1977*; *Schuknecht & Donovan, 1986*). Demographic data are shown in Table 1. The vestibular involvement was relatively higher in our experimental group compared to previous reports (*Moskowitz, Lee & Smith, 1984*; *Park, Jung & Rhee, 2001*; *Shaia & Sheehy, 1976*). This could be related to subjects who were enrolled in this study, since the MRI is not a routine study
**Table 1  Demographic data of patients in present study.**

| Demographics | |
| --- | --- |
| Mean age (±Standard deviation) | 45.0 (±15.3) |
| Hospital day of MR imaging (±Standard deviation) | 3.7 (±2.2) |
| Gender | M 59% vs F 41% |
| Patients with accompanying symptom | |
| Vertigo (%) | 31 (63.3) |
| Tinnitus (%) | 31 (63.3) |

for hearing loss in our health system. Patients with MRI imaging of the IAC with no signs of vestibular schwannoma were included in the study. Patients were classified according to the anatomical location of the AICA and cranial nerves within the IAC. Two established classification methods designed by Chavda (*McDermott et al., 2003*) and Gorrie (*Gorrie et al., 2010*) were used. Pure tone threshold at four different frequencies (0.5, 1, 4, and 8 kHz), at two different time points (at time of initial presentation, and three months after initial treatment), were analyzed. All patients were given anti-viral agent, systemic high dose steroid therapy (48 mg at day time, 12 mg at night time, total 60 mg of methylprednisolone for 7 days) and non-systemic steroid responsive (mean recovery average less than 10 dB HL) subjects had additional intra-tympanic steroid injections. We compared the affected and unaffected ears with two different classification systems, and investigated whether there were differences in hearing recovery and symptoms. Hearing improvement was assessed by Siegel's criteria (average of 0.5, 1, 2 and 4 kHz) (*Siegel, 1975*) and measuring the first threshold shift between time points at each frequency, and documenting the proportion of patients with improved hearing (>10 dB HL) at each separate frequency.

## MRI protocol

All MRI test were conducted using a 3 T scanner (signa HDxt, GE Medical system, Milwaukee, WI) with an eight channel head coil. Among the routine IAC MR imaging protocol, 3D T2 VISTA images were selected for analyzing the anatomical configurations of IAC vessel and cranial nerves. Two classification systems were adopted. The first was the Chavda classification published by *McDermott et al. (2003)*. This system classifies AICA types as follows: type 1 is an AICA loop within the CPA but outside the IAC; type 2 is an AICA loop extending into the IAC but is less than 50% the length of the IAC; type 3 is an AICA loop with greater than 50% extension into the IAC (Fig. 1). The second classification system used was the Gorrie type, which is based on the amount of contact of between the AICA and adjacent cranial nerves. Type 1 is an AICA loop without contact to adjacent nerves; type 2 is an AICA loop that runs adjacent to the nerves; type 3 is an AICA loop that physically displaces the 8th cranial nerve; type 4 is an AICA loop that courses between the 7th and 8th cranial nerves (Fig. 2). All MR images were analyzed and classified by a radiologist who is co-author of our manuscript (SYK).

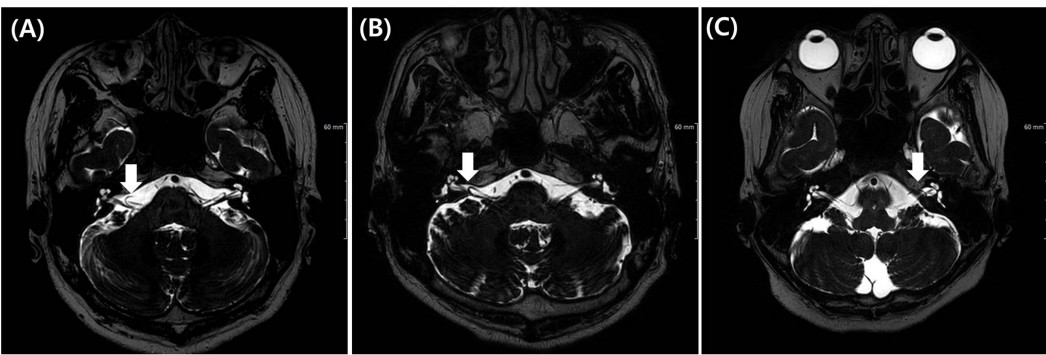

**Figure 1  Chavda classification of AICA loop.** (A) AICA loop (arrow) observed in cerbellopontine angle (CPA) outside the internal auditory canal (IAC) which is type 1. (B) Type 2 in which AICA loop (arrow) is occupying no more than 50% of IAC. (C) Type 3 in which AICA loop (arrow) leaches more than 50% of total length of IAC.

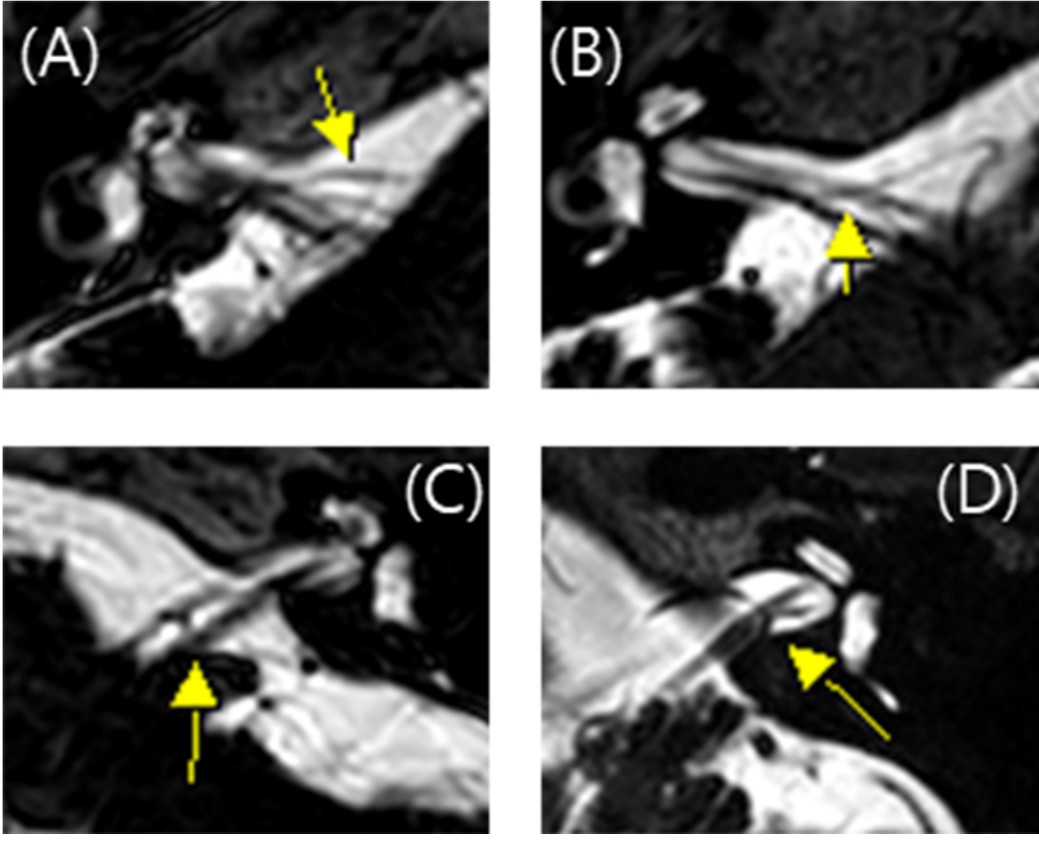

**Figure 2  Gorrie classification of AICA loop.** (A) AICA loop (arrow) running separate from cranial nerve which is type 1. (B) Type 2 in which the AICA loop (arrow) is running adjacent to the cranial nerve. (C) Type 3 in which the AICA loop (arrow) deflects the 8th cranial neve and (D) Type 4 in which the AICA loop (arrow) runs between the 7th and 8th cranial nerve.

**Table 2  Chavda type and Gorrie type distributions in ISSNHL and contralateral ear.**

|  | Ipsilateral ear (%) | Contralateral ear (%) | *p*-value |
|---|---|---|---|
| Chavda type I | 25 (51.0%) | 29 (59.2%) | |
| Chavda type II | 22 (44.9%) | 19 (38.8%) | 0.651 |
| Chavda type III | 2 (4.1%) | 1 (2.0%) | |
| Gorrie type I | 7 (14.3%) | 10 (20.4%) | |
| Gorrie type II | 9 (18.4%) | 8 (16.3%) | |
| Gorrie type III | 29 (59.2%) | 24 (49.0%) | 0.598 |
| Gorrie type IV | 4 (8.1%) | 7 (14.3%) | |

## Statistical analysis

All data were analyzed by GraphPad Prism (GraphPad Software, La Jolla, CA, USA) or SPSS (IBM SPSS statistics, Armonk, NY, USA) software. A Shapiro–Wilk normality test was used to determine whether the data were parametric or non-parametric. Significant differences between groups were statistically analyzed using *t*-test in cases of a parametric distribution, and Mann–Whitney *U* test in cases of a nonparametric distribution. Fischer's exact test was used for the cross-table analysis. A *p*-value less than 0.05 was considered statistically significant.

## RESULTS

Pure tone averages of ears with ISSNHL were 73.6, 76.9, 78.1, and 77.1 at 0.5, 1, 4 and 8 kHz respectively, and those of the contralateral side were 11.5, 12.3, 23.3, and 29.7 at 0.5, 1, 4 and 8 kHz respectively. The average threshold shift of the contralateral ear at each frequency was no greater than 30 dB HL, suggesting near normal hearing function. We compared the anatomical position of the AICA loop between ears with ISSNHL and the unaffected contralateral ear. The types of anatomical variations of the AICA were not different between the affected side and the contralateral side (Table 2) ($p > 0.05$, Fisher's exact test). With regard to the Chavda classification, Chavda type I was the most common, followed by type II, and type III. For the Gorrie classification, the Gorrie type III was the most common (Table 2).

We also analyzed symptoms such as vertigo and tinnitus. The relationship between AICA and symptoms were classified as Tables 3 and 4. As a result, the anatomic variations of AICA was not different according to vertigo and tinnitus, respectively ($p > 0.05$).

We compared the threshold shift from the start of the treatment and at 3 months. In all four frequencies, Chavda type 1 showed the largest threshold improvement but was not statistically significant (Fig. 3) (500 Hz: Kruskal–Wallis test, KW statistics $= 4.091$, $p = 0.13$; 1 kHz: Kruskal–Wallis test, KW statistics $= 4.719$, $p = 0.09$; 4 kHz: Kruskal–Wallis test, KW statistics $= 4.789$, $p = 0.09$; 8 kHz: Kruskal–Wallis test, KW statistics 3.336, $p = 0.19$, mean hearing level: Kruskal–Wallis test, KW statistics 3.381, $p = 0.18$). At lower frequencies (500 Hz, 1 kHz), hearing improvements were found in type 2 and type 3 Gorrie configurations. Higher frequencies (4 kHz, 8 kHz) did not yield any significant differences in hearing improvements, with a Gorrie type 4 at 4 kHz improving the least. These differences were

**Table 3  Chavda type and Gorrie type distributions in ISSNHL with vertigo and without vertigo.**

|  | With vertigo (%) | Without vertigo (%) | *p*-value |
|---|---|---|---|
| Chavda type I | 15 (48.4%) | 10 (55.6%) |  |
| Chavda type II | 14 (45.2%) | 8 (44.4%) | 0.528 |
| Chavda type III | 2 (6.4%) | 0 (0.0%) |  |
| Gorrie type I | 5 (16.1%) | 2 (11.1%) |  |
| Gorrie type II | 6 (19.4%) | 3 (16.7%) |  |
| Gorrie type III | 17 (54.8%) | 12 (66.7%) | 0.861 |
| Gorrie type IV | 3 (9.7%) | 1 (5.5%) |  |

**Table 4  Chavda type and Gorrie type distributions in ISSNHL with and without tinnitus.**

|  | With tinnitus (%) | Without tinnitus (%) | *p*-value |
|---|---|---|---|
| Chavda type I | 21 (55.3%) | 4 (36.4%) |  |
| Chavda type II | 16 (42.1%) | 6 (54.5%) | 0.414 |
| Chavda type III | 1 (2.6%) | 1 (9.1%) |  |
| Gorrie type I | 5 (13.2%) | 2 (18.2%) |  |
| Gorrie type II | 9 (23.7%) | 0 (0.0%) |  |
| Gorrie type III | 22 (57.9%) | 7 (63.6%) | 0.208 |
| Gorrie type IV | 2 (5.2%) | 2 (18.2%) |  |

not statistically significant (Fig. 4) (500 Hz: Kruskal–Wallis test, KW statistics = 2.770, $p = 0.43$; 1 kHz: Kruskal–Wallis test, KW statistics = 3.811, $p = 0.28$; 4 kHz: Kruskal–Wallis test, KW statistics = 3.609, $p = 0.31$; 8 kHz: Kruskal–Wallis test, KW statistics 0.2900, $p = 0.96$, mean hearing level: Kruskal–Wallis test, KW statistics 3.277, $p = 0.35$). According to the classification of Siegel's criteria, it was found that improved groups, from slight to complete recovery, showed higher Chavda type 1 proportion ($>50\%$) compared to no recovery (Chavda type $1 < 30\%$) but it failed to reveal statistical significance (Fischer exact test, $p = 0.50$). As hearing improved across both classifications, all Chavda types had a significant difference at 1 kHz and 4 kHz (Fischer exact test, 1 kHz: $p = 0.03$, 4 kHz: $p = 0.01$). There was no statistical significance at other frequencies in Chavda or Gorrie type configurations (Chavda type; Fischer exact test, 500 Hz: $p = 0.17$, 8 kHz: $p = 0.14$) (Gorrie type; Fischer exact test, 500 Hz: $p = 0.47$, 1 kHz: $p = 0.36$, 4 kHz: $p = 0.11$, 8 kHz: $p = 0.92$) (Tables 5 and 6).

## DISCUSSION

In the present study, there were no differences in AICA types in ears with or without ISSNHL. Patients who had combined symptoms such as tinnitus and vertigo did not show a different distribution of AICA type compared to patients without symptoms. This suggests that AICA type does not affect the incidence of ISSNHL and concurrent symptoms. There were some differences in quantitative hearing improvement between types, although without statistical significance (*p* values between 0.09 and 0.13). At two frequencies, 1 and 4 kHz, there was a difference in Chavda types between patients who experienced hearing

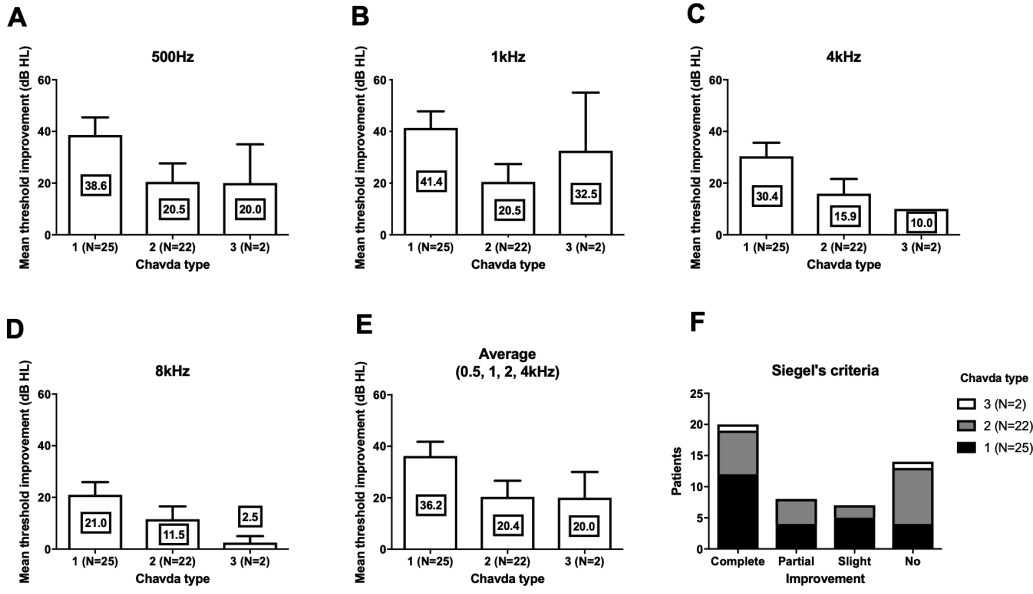

**Figure 3 Hearing threshold improvement across different Chavda types.** At all frequencies (A–D), highest threshold improvements were observed in Chavda type 1. At lowest frequency (500 Hz, (A)), similar hearing improvement were observed in Chavda type 2 and 3. At 1 kHz, Chavda type 3 showed higher improvement (B). In contrast, both in 4 kHz (C) and 8 kHz (D), Chavda type 2 showed higher improvement. Average of 0.5, 1, 2 and 4 kHz was compared and revealed high improvement in Chavda type 1 (E). In Siegel's criteria (F), proportion of Chavda type 1 was small in no improvement group. Nevertheless, all of these comparisons among Chavda types failed to reveal statistical significance (see the results for detailed statistics). The number in center of each bar means each mean hearing threshold improvement (dB HL). Error bar indicates standard deviation.

improvement and patients who did not. In groups that had improvements in hearing, we found a higher proportion of Chavda type 1 configurations (AICA locating outside the IAC). These results suggest that the anatomic location of the AICA loop may help prognosticate hearing outcomes in ISSNHL patients.

Currently, there is no clear etiology of AICA loop formation and anatomical variances seen in AICA positions. Hypotheses include senile elongation of the artery, arteriosclerosis, and arachnoid adhesions between nerves and vessels (*Applebaum & Valvassori, 1984*). The prevalence of AICA loops inside the IAC is thought to be approximately 13 to 40% in cadaveric dissections (*Mazzoni & Hansen, 1970*; *Reisser & Schuknecht, 1991*) and 14 to 34% in imaging studies using MRI (*De Carpentier et al., 1996*; *Sirikci et al., 2005*). In the current study the percentage of patients found to have a Chavda type 1 configuration was 51% (59% in control side), which is a smaller than results from previous studies (60 to 87%). This difference may be attributable to variations in study number, age, or sex. Future studies that match healthy controls with subjects may be useful in further evaluating the relationship between ISSNHL incidence and AICA loop location.

Microvascular compression is thought to be responsible for certain cases of hearing loss, tinnitus, vertigo, and hemifacial spasm (*Jannetta, 1980*). The results of our study are in line with previous findings, with no differences being found in the relationship between AICA
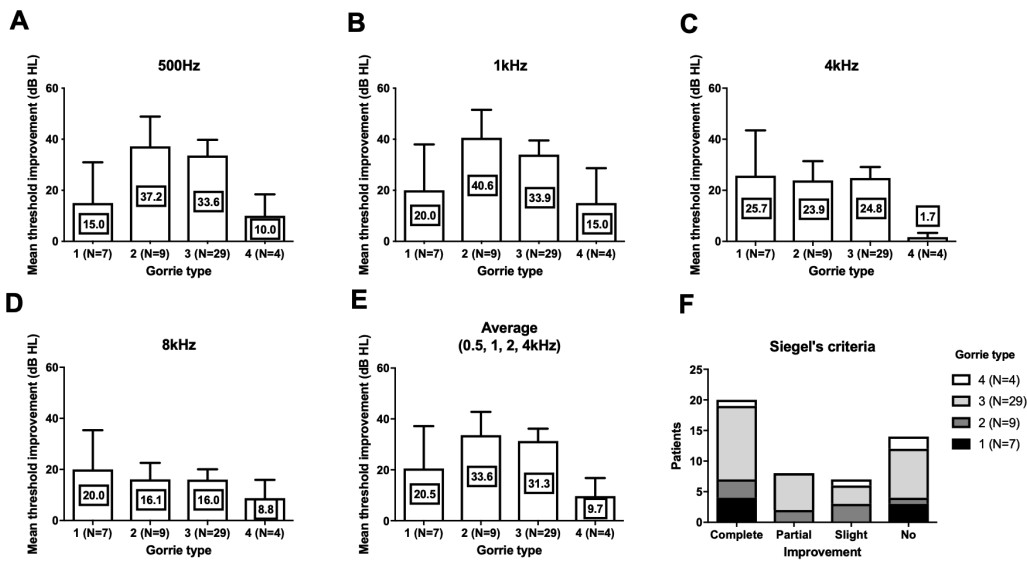

**Figure 4  Hearing threshold improvement across different Gorrie types.** At 500 Hz (A) and 1 kHz (B), Gorrie type 2 and 3 showed higher threshold improvement. At 4 kHz (C) Gorrie type 4 showed least improvement of hearing. At 8 kHz (D), threshold improvement across Gorrie types were similar. Average of 0.5, 1, 2 and 4 kHz was compared and showed similar Gorrie type distribution to 500 Hz and 1 kHz (E). In Siegel's criteria (F), proportion of Gorrie types was not different among groups. Nevertheless, all of these comparisons among Gorrie types failed to reveal statistical significance (see the results for detailed statistics). The number in center of each bar means each mean hearing threshold improvement (dB HL). Error bar indicates standard deviation.

**Table 5  Chavda type proportion of hearing improved cases at each frequencies.**

|  | Cahvada type I (n = 25) | Cahvada type 2 (n = 22) | Cahvada type 3 (n = 2) | p-value |
|---|---|---|---|---|
| 500 Hz (%) | 17 (68.0%) | 9 (40.9%) | 1 (50.0%) | 0.174 |
| **1 kHz (%)** | **19 (76.0%)** | **9 (40.9%)** | **1 (50.0%)** | **0.049** |
| **4 kHz (%)** | **19 (76.0%)** | **9 (40.9%)** | **0 (0.0%)** | **0.013** |
| 8 kHz (%) | 13 (52.0%) | 6 (27.2%) | 0 (0.0%) | 0.114 |

Notes.
    Bold: statistically significant.

**Table 6  Gorrie type proportion of hearing improved cases at each frequencies.**

|  | Gorrie type 1 (n = 7) | Gorrie type 2 (n = 9) | Gorrie type 3 (n = 29) | Gorrie type 4 (n = 4) | p-value |
|---|---|---|---|---|---|
| 500 Hz (%) | 3 (42.8%) | 6 (66.6%) | 17 (58.6%) | 1 (25.0%) | 0.472 |
| 1 kHz (%) | 4 (57.2%) | 7 (77.7%) | 17 (58.6%) | 1 (25.0%) | 0.356 |
| 4 kHz (%) | 4 (57.2%) | 6 (66.6%) | 18 (62.0%) | 0 (0.0%) | 0.114 |
| 8 kHz (%) | 3 (42.8%) | 4 (44.4%) | 11 (37.9%) | 1 (25.0%) | 0.919 |

loop distribution and symptomatic/non-symptomatic patients (*De Carpentier et al., 1996*; *Gorrie et al., 2010*; *Sirikci et al., 2005*). We also did not find a higher incidence of Gorrie type 3 and 4 configurations in patients with tinnitus, thereby decreasing the likelihood that symptoms could be due to contact of the AICA loop with the cochlear nerve. However, given the low percentage of pulsatile tinnitus compared to subjective tinnitus patients in our study, we believe that our current data are insufficient to comment further on the previously studied (*Chadha & Weiner, 2008*; *De Ridder et al., 2005*) relationship between tinnitus (vascular and non-vascular) and AICA loop position in ISSNHL patients.

Quantitative hearing improvement failed to reveal significant differences, although the patient group that had hearing improvement showed different Chavda type proportions. This finding may be due to chance or to low sample sizes. Nevertheless, a plausible explanation for the improved prognosis of Chavda type 1 configurations is necessary. Among many possible etiologies of ISSNHL, two most highly adopted theories are viral inflammation and ischemia. Inflammation of cochlear nerve can be due to a variety of infectious causes, and results in reversible axonal swelling and degeneration. Most cases retain a good prognosis with appropriate therapeutic management with appropriate management such as steroid (to reduced secondary damage due to edema) and antiviral agent. In contrast, ischemic attack results in irreversible cochlear damage due to sensorineural cell death, (*Izumikawa et al., 2005*) leading to poorer outcomes. Given that the pathophysiology of ISSNL is thought to be multifactorial, the group that experienced less hearing improvements (Chavda type 2 or 3) may have had a higher rate of vascular etiologies compared to patients with a Chavda type 1 configuration. Unlike Gorrie classification which is classified by the contact of vessel and nerves, Chavda classification divides groups by distance of AICA loop and IAC. In case of Chavda type 2 and 3, AICA loop is located within the narrow IAC which in many case leads to smaller diameter of AICA loop and sudden rotation. On the contrary, in the Chavda type 1, AICA loop is formed in CPA outside of IAC which has relatively larger space, abrupt rotation of the loop is not always necessary in this case. The turbulence, which is relevant factor in thrombus formation within the AICA loop (*Bluestein et al., 1997*; *Deusebio et al., 2014*), may occur at a higher rate in Chavda type 2 or 3 patients (small diameter IAC loop and narrow space), and supports the notion that labyrinthine artery ischemia is more common in these patients, resulting in a higher rate of no-improvement hearing outcome from ISSNHL. Furthermore, outcome variabilities among different frequencies are another evidence to speculate the pathophysiology. The statistical group difference was observed in 1, 4 kHz not in 500 Hz and 8 kHz. The possible reason for no group difference in 8 kHz might be related to small hearing improvements, as observed in Figs. 3 and 4. On the other hand, 500 Hz hearing improvement was relatively similar to other frequencies (Figs. 3 and 4) and there should be an alternate plausible theory. Considering the tonotopicity of cochlear nerve (*Muller, 1991*), the frequencies which showed group differences (better improvement in Chavda type 1) would be the peripheral part of cochlear nerve (except the highest frequency). In treatment responsive population which had high proportion of Chavda type 1 (could be viral origin); damage of cochlear nerve fiber could be focused

in peripheral axons which are close to the nerve sheath and susceptible to the pressure increase.

On the other hand, it is possible to argue that in Chavda type 1 response to treatment was better compared to other types. Systemically delivered steroid and antiviral agent could reach the target organ faster in case of lesser complicated anatomical positioning of AICA, such as Chavda type 1, studies comparing outcomes and AICA classifications of the local and systemic treatment would help better understanding considering this point.

## CONCLUSION

Anatomical variances in AICA loop position did not affect the incidence of ISSNHL or co-morbid symptoms. In contrast, comparisons between groups with improvements in hearing and those without revealed that a higher proportion of Chavda type 1 (AICA outside IAC) patients had better prognostic outcomes.

### Funding
This research was supported by a grant of the Korea Health Technology R&D Project through the Korea Health Industry Development Institute (KHIDI), funded by the Ministry of health & Welfare, Republic of Korea (grant number : HI15C1524). The funders had no role in study design, data collection and analysis, decision to publish, or preparation of the manuscript.

### Grant Disclosures
The following grant information was disclosed by the authors:
Korea Health Industry Development Institute (KHIDI).
Ministry of health & Welfare, Republic of Korea: HI15C1524.

### Competing Interests
The authors declare there are no competing interests.

### Author Contributions
- Sang Hyub Kim and Yeo Rim Ju performed the experiments, analyzed the data, contributed reagents/materials/analysis tools, prepared figures and/or tables.
- Ji Eun Choi and Jae Yun Jung conceived and designed the experiments, authored or reviewed drafts of the paper.
- Sang Yoon Kim performed the experiments, analyzed the data, contributed reagents/materials/analysis tools.
- Min Young Lee conceived and designed the experiments, analyzed the data, contributed reagents/materials/analysis tools, prepared figures and/or tables, approved the final draft.

### Human Ethics
The following information was supplied relating to ethical approvals (i.e., approving body and any reference numbers):

This study has been approved by the Ethical Committee of Faculty of Medicine, Dankook University Hospital (Ethical Application Ref: 2017-08-003).

## Data Availability

The raw measurements are available in File S1. The raw data includes patient id and other information.

## Supplemental Information

Supplemental information for this article can be found online at http://dx.doi.org/10.7717/peerj.6582#supplemental-information.

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
