# Peer review of "Anatomical location of AICA loop in CPA as a prognostic factor for ISSNHL"

_PeerJ, doi:10.7717/peerj.6582_

## Round 0.1 · original submission · Minor Revisions

Dear Authors,

Please revise your manuscript according to the comments if the peer reviewers soonest so as it can be re-reviewed by them again to improve the quality of the manuscript.

Reviewer 1 ·

Basic reporting

no comment

Experimental design

no comment

Validity of the findings

no comment

Additional comments

It is a good manuscript focus-analysing only on ISSNHL group. The methods, results and discussion are scientifically comprehensible and have added value to the current literature.

Reviewer 2 ·

Basic reporting

In my opinion, this is a well-written article with a clear and professional English language. The literature is well referenced and relevant. The structure is conformed to PeerJ standard.
About the figure, I have some opinions :
1. Figure 3 is unnecessary since the hearing thresholds of affected and nonaffected ears are presented clearly in the Results section.
2. It is better to clearly mention the definition of the y-axis and the error bar in the figure legend of figure 4 and 5. For example, mean (or average) hearing threshold improvement with standard deviation (or error)
3. In Figure 4D, the y-axis of dBHL shall be dB HL.
In the table, Chavda classification sometimes is typed as “Cahvada”.
The statement of asterisk in statistical analysis (*p<0.05, **p<0.01, and ***p<0.001 ) is unnecessary because I do not see any asterisk in results section and figures/tables.

Experimental design

The original primary research is within Scope of the journal (Medical science). Research question well is defined, relevant and meaningful. However, I have some comments about experimental designs :
1. The authors were suggested to demonstrate more the meaning of two different classifications of Chavda and Goirred on their study and their hypothesis. Do they hypothesize that the distance of AICA and IAC correlate to the occurrence and prognosis of ISSNHL?
2. Did all of the subjects treat with high dose steroid only? Are there any subjected who have ever receive intratympanic injections of steroid? Since salvage therapies were important factors that affect the prognosis of ISSNHL, the authors were suggested to mention their protocol to treat ISSNHL more detailedly.
3. The authors only define hearing improvement as more than 10 dB HL at each frequency. However, one who can improve 11 dB HL at 1kHz may not improve at 4kHz. Probably another grading system such as Siegel’s criteria may be better to reflect the hearing outcome of ISSNHL patients.

Validity of the findings

The author speculated that the group that experienced hearing improvements (Chavda type 1) might have had a higher rate of inflammatory/infectious etiologies compared to patients with a Chavda type 2 or 3 configurations so that patients with Chavda can respond better to steroid therapy. However, if it is the case, patients with Gorrie type 1 will have a better prognosis, which is not shown in the results. In addition, why the improvement of Chavda type 1 is only found over 1 KHz and 4 KHz needs further elucidation.

Additional comments

It is an interesting article that provides the readers to think about other possible etiologies of ISSNHL as well as factors that affect the hearing outcomes of ISSNHL. However, more discussions were suggested to speculate their conclusion that the anatomic location of the AICA loop may help prognosticate hearing outcomes in ISSNHL patients.

·

Basic reporting

Line 85: please check the amount of methylprednisolone. Because, MPD is usually used as 0.8mg/kg/d. Or please describe your steroid protocol.

Experimental design

Line 83
it is interesting that PTA was calculated by the average of 0.5, 1,4,8 kHz. And it is different from other ISSNHL studies. Please consider including results of other frequencies or focus on the mean hearing level instead.

Line 112-113:
As described earlier, reporting PTA at 0.5, 1, 4, 8 khz separately is a little bit unsual. Please consider reporting of PTA at all frequencies from 250 Hz to 8 kHz. In addition, the authors are asked to add the mean hearing level of both sides.

Line 116-122
1. The authors are asked to write results of the Chavda & the Gorrie in next paragraph.
2. Table 2 should be mentioned in the beginning of the new paragraph like this.

“The types of anatomical variations of the AICA were not different between the affected side and the contralateral side (Table 2) (p>0.05, Fisher’s exact test). With regard to the Chavda classification, Chavda type I was the most common, followed by type II, and type III. For the Gorrie classification, the Gorrie type III was the most common (Table 2).”

3. Please delete the following sentence. “Using the Chavda classification, both ears showed a high proportion of type 1 (greater than 50%), with type 2 having the next highest proportion. “. It’s redundant.

4. Table 2: The Cahvada may be the typo of Chavda. Please correct the typo.

Line 123-132.
Please delete the following sentence starting from Patients were separated into … . It’s also redundant. To clarify, the authors should simplify the results of this paragragh like this.
“The relationship between AICA and symptoms were classified as Table 3 and 4. As a result, the anatomic variations of AICA was not different according to vertigo and tinnitus, respectively (p>0.05).”

Line 133-146

1. With regard to figure 4 and 5, Please add the mean hearing level in the center of bars at all figures.
2. The authors are asked to add the mean hearing level and the hearing gain according to each classification, respectively. Figures may be better to express the results more efficiently and the numbers should be included in the figures.
3. Please consider including results of other frequencies such as 250 Hz, 2 kHz, 3 kHz. Or, concentrate on the mean hearing level according to each classifications.



Table 1:
The proportion of vertigo seems to be higher than other studies. The usual co-incidence of vertigo was reported to be about 30%. Please describe what made this in the discussion section.

Validity of the findings

Line 201-202
Please delete the final two sentences staring from limitations -, and We highly recommend…

By this, your final conclusion will be more clear.

Additional comments

In this study, the authors analyzed the relationship between the AICA and the various prognotisic factors of the sudden hearing loss. As a result, they found that hearing outcome may be influenced by AICA loop to a small extent.


The authors are asked to add more descriptions why the location of AICA matters in the clinial course of idiopathic sudden hearing loss. The authors described that the anatomical variations of the AICA results in the ISSNHL. However, it seems to be too strong statement. Although labyrinthine artery is a branch of AICA, there is no consensus what proportions of sudden hearing loss is caused by problems in the vessels. And the authors focused on the vascular loop, not a labyrinthine artery.

Reviewer 4 ·

Basic reporting

This article is written clearly in unambiguous English throughout. It adheres to acceptable formatting standards and the figures and tables are relevant to the study. Relevant permission has also been obtained from the local review board allowing this work to be done. The literature on the subject is also relevant and up to date
The authors have set out to determine the relationship between the shape and location of the AICA and its correlation with the clinical features and prognosis of ISSNHL. This is a simple study that was performed by retrospectively reviewing data of patients who presented over a two year period to their centre.
In the introduction, the authors have made adequate effort to introduce the studied parameters and the relevance of their hypothesis. The methodology including imaging and clinical parameters evaluated are clearly described. The raw data of patients have also been shared.

Experimental design

The experiment is original work that has been appropriately designed to answer the research question. The results are acceptably described in the text.The statistical tests are also appropriately applied The Tables however could be improved. In Table 1 depicting demography, it would be better to state the category of demographic data being represented rather than listing all different parameters one after another. Example Gender : Male x% vs Female y% . Statistical test applied and the p value findings should also be added to the tables for ease of reading rather than having to go back to the text each time. (For Tables 2 – 6). The Figure depicting the Chavda and Gorrie MRI classifications are appropriate.

Validity of the findings

Despite their being studies to correlate the anatomy of the AICA with audiovestibular symptoms, the authors are the first to study this anatomy in patients with ISSNHL. The work has noted no statistically significant findings indicating that their might be any relationship with the AICA anatomy and incidence of ISSNHL. The demonstrated improvement in hearing and its relationship to the anatomical subtypes also did not reveal significance statistically.A larger small sample size is needed to confirm this point.

Additional comments

Overall, this is a well conducted simple retrospective effort to study AICA anatomy and its correlation with ISSNHL. The methodology is appropriate, however the results tables should be appended to better represent the data in terms of statsitical tests applied.

---

## Round 0.2 · Minor Revisions

Dear Authors,

Please revise the statistical components that are highlighted :

1.I still do not see the definition of error bar shown in figure 4 and 5. Are they standard deviation or standard error ?

2.The statement of asterisk in statistical analysis (*p<0.05, **p<0.01, and ***p<0.001 ) is suggested to be deleted since there are no asterisks shown in results section and figures/tables.

Thank you

Reviewer 1 ·

Basic reporting

Clear

Experimental design

Original

Validity of the findings

Valid

Additional comments

accepted as commented by me before.

Reviewer 2 ·

Basic reporting

no comment

Experimental design

no comment

Validity of the findings

no comment

Additional comments

The authors revised their manuscript well. Minor revisions are suggested as below:

1. I still do not see the definition of error bar shown in figure 4 and 5. Are they standard deviation or standard error ?
2. The statement of asterisk in statistical analysis (*p<0.05, **p<0.01, and ***p<0.001 ) is suggested to be deleted since there are no asterisks shown in results section and figures/tables.

·

Basic reporting

no comment

Experimental design

no comment

Validity of the findings

no comment

Additional comments

Thank you for hard work.

---

## Round 0.3 · accepted · Accept

Dear Author,

The Peer Reviewer is extremely happy that the re-revisions to the manuscript have been made satisfactorily leading to an acceptance for publication in PeerJ.

# Reviewer 2 ·

Basic reporting

no comment

Experimental design

no comment

Validity of the findings

no comment

Additional comments

The authors revised their manuscript as my suggestion. Thank you for hard work.